# The chromatin landscape of primary synovial sarcoma organoids is linked to specific epigenetic mechanisms and dependencies

Gaylor Boulay[3,4,*], Luisa Cironi[1,2,*], Sara P Garcia[3,†], Shruthi Rengarajan[3,†], Yu-Hang Xing[3], Lukuo Lee[3], Mary E Awad[3], Beverly Naigles[3], Sowmya Iyer[3], Liliane C Broye[1,2], Tugba Keskin[1,2], Alexandra Cauderay[1,3], Carlo Fusco[1,2], Igor Letovanec[1], Ivan Chebib[3], Petur Gunnalugur Nielsen[3], Stéphane Tercier[6], Stéphane Cherix[5], Tu Nguyen-Ngoc[7], Gregory Cote[8], Edwin Choy[8], Paolo Provero[9,10], Mario L Suvà[3,4], Miguel N Rivera[3,4], Ivan Stamenkovic[1,2], Nicolò Riggi[1,2]

Synovial sarcoma (SyS) is an aggressive mesenchymal malignancy invariably associated with the chromosomal translocation t(X:18; p11:q11), which results in the in-frame fusion of the BAF complex gene *SS18* to one of three *SSX* genes. Fusion of SS18 to SSX generates an aberrant transcriptional regulator, which, in permissive cells, drives tumor development by initiating major chromatin remodeling events that disrupt the balance between BAF-mediated gene activation and polycomb-dependent repression. Here, we developed SyS organoids and performed genome-wide epigenomic profiling of these models and mesenchymal precursors to define SyS-specific chromatin remodeling mechanisms and dependencies. We show that SS18-SSX induces broad BAF domains at its binding sites, which oppose polycomb repressor complex (PRC) 2 activity, while facilitating recruitment of a non-canonical (nc)PRC1 variant. Along with the uncoupling of polycomb complexes, we observed H3K27me3 eviction, H2AK119ub deposition and the establishment of de novo active regulatory elements that drive SyS identity. These alterations are completely reversible upon SS18-SSX depletion and are associated with vulnerability to USP7 loss, a core member of ncPRC1.1. Using the power of primary tumor organoids, our work helps define the mechanisms of epigenetic dysregulation on which SyS cells are dependent.

## Introduction

Synovial sarcoma (SyS) is the second most common soft tissue malignancy in the adolescent and young adult population, in which it comprises 10–20% of all soft tissue sarcomas (Nielsen et al, 2015). Its defining genetic feature is the chromosomal translocation t(X:18; p11: q11), which generates fusions between the nearly entire coding sequence of the *SS18* gene and a portion of an *SSX* gene family member (*SSX1* and *SSX2* being the two most commonly implicated [Nielsen et al, 2015; Riggi et al, 2018]). As is often observed in pediatric malignancies, SyS displays few genetic mutations other than the chromosomal translocation itself, suggesting that SS18-SSX bears the prime responsibility for SyS pathogenesis. Accordingly, a transgenic mouse model of SyS was generated by expressing human *SS18-SSX* in an early Myf5+ murine myoblast population (Haldar et al, 2007). Whereas this model pointed to Myf5+ myoblasts as a potential cell of origin of SyS, experiments using conditional expression of SS18-SSX produced tumors in atypical anatomic locations, suggesting that other cell types may also be permissive for SS18-SSX–mediated transformation. Consistent with this notion, mouse and human mesenchymal stem cells (MSCs) have been shown to acquire a gene expression profile that resembles that of primary human SyS upon expression of SS18-SSX (Cironi et al, 2009, 2016).

The oncogenic properties of SS18-SSX have been linked to its ability to interact with proteins that control epigenetic states (Thaete

[1]Institute of Pathology, Centre Hospitalier Universitaire Vaudois, Faculty of Biology and Medicine, University of Lausanne, Lausanne, Switzerland    [2]Swiss Cancer Center Leman, Centre Hospitalier Universitaire Vaudois, Faculty of Biology and Medicine, University of Lausanne, Lausanne, Switzerland    [3]Department of Pathology and Center for Cancer Research, Massachusetts General Hospital and Harvard Medical School, Boston, MA, USA    [4]Broad Institute of Harvard and MIT, Cambridge, MA, USA    [5]Department of Orthopedics, Faculty of Biology and Medicine, Centre Hospitalier Universitaire Vaudois, Lausanne, Switzerland    [6]Department of Woman-Mother Child, Centre Hospitalier Universitaire Vaudois, Faculty of Biology and Medicine, University of Lausanne, Lausanne, Switzerland    [7]Department of Oncology, Centre Hospitalier Universitaire Vaudois, Faculty of Biology and Medicine, University of Lausanne, Lausanne, Switzerland    [8]Division of Hematology and Oncology, Department of Medicine, Massachusetts General Hospital, Boston, MA, USA    [9]Center for Translational Genomics and Bioinformatics, San Raffaele Scientific Institute, Milan, Italy    [10]Department of Molecular Biotechnology and Health Sciences, University of Turin, Turin, Italy

Correspondence: Ivan.Stamenkovic@chuv.ch; nicolo.riggi@chuv.ch
*Gaylor Boulay and Luisa Cironi contributed equally to this work
†Sara P Garcia and Shruthi Rengarajan contributed equally to this work
Miguel N Rivera, Ivan Stamenkovic, and Nicolo Riggi are senior authors

et al, 1999; Nagai et al, 2001). SS18 is a component of the mammalian chromatin remodeling complex switch/sucrose non-fermentable (SWI/SNF also known as BRG1/BRM-associated factor, BAF) (Wang et al, 1996a, 1996b; Kadoch & Crabtree, 2013), which facilitates transcription by increasing chromatin accessibility at promoter and enhancer regions. In contrast, SSX1/2 belongs to a family of transcriptional repressors that co-localize with polycomb group proteins (PcG), including BMI1 and RING1B (Hale et al, 2019; Johansen & Gjerstorff, 2020). The two constituents of the fusion protein are thus associated with opposing effects on chromatin regulation and the integration of their activities appears to provide the key toward establishing the oncogenic signaling network that generates SyS. Biochemical studies have shown that SS18-SSX replaces wt SS18 in the BAF complex and alters its composition by ejecting the tumor suppressor SNF5/BAF47 (Kadoch & Crabtree, 2013). The resulting effect is the redirection of the newly constituted BAF complexes from enhancers to PcG domains to relieve polycomb repressor complex (PRC)2–mediated repression of bivalent target promoters (McBride et al, 2018). Recent functional genomic experiments have also uncovered physical interaction between SS18-SSX and the histone demethylase KDM2B, a component of a non-canonical polycomb repressive complex 1 (ncPRC1.1), which leads to aberrant reactivation and expression of PcG target genes that are required for transformation (Banito et al, 2018a). An additional repressive role for SS18-SSX, which leads to PRC2 recruitment to ATF2 target genes, has also been proposed (Su et al, 2012).

Given the importance of epigenetic events as drivers of its pathogenesis, chromatin regulators constitute potentially attractive therapeutic targets for SyS. Accordingly, recent studies suggest that small-molecule–mediated degradation of the BAF component BRD9 can reverse oncogenic gene regulation in SyS (Brien et al, 2018; Michel et al, 2018). However, the development of new anticancer therapies is hindered by the paucity of reliable pre-clinical models. Patient-derived tumor organoids have emerged as a powerful system for modeling key biological properties of human carcinomas and their response to therapy (Drost & Clevers, 2018). Whereas organoids from human sarcomas have generally been challenging to establish, we have shown that 3D organoids derived from primary Ewing sarcomas recapitulate key native tumor features more closely than cell lines maintained in 2D culture, including inter- and intra-tumor heterogeneity, cellular plasticity and response to therapy (De Vito et al, 2012; Cornaz-Buros et al, 2014).

In the present study, we sought to generate primary models of SyS that capture its regulatory landscape and provide insight into its driver mechanisms and epigenetic dependencies. We therefore produced a set of patient-derived SyS organoids and assessed their genome-wide regulatory features through chromatin profiling. Our observations reveal that expression of SS18-SSX in both organoids and mesenchymal precursor cells is associated with a distinctive distribution of the BAF complex and the uncoupling of PRC1 and PRC2. Together, these events initiate a SyS-specific oncogenic program that is completely reversible upon SS18-SSX silencing, suggesting that a pharmacological approach aimed at epigenetic regulatory events may offer realistic therapeutic opportunities in established tumors. Accordingly, we found SyS cells to be highly vulnerable to the depletion of ubiquitin-specific protease 7 (USP7), a member of ncPRC1 involved in maintaining SyS cell proliferation. Taken together, our results show that the chromatin landscape of Sys organoids not only reflects the key pathogenic mechanisms that lead to SyS but is also linked to specific epigenetic dependencies.

# Results

## Primary synovial sarcoma organoids have a distinctive pattern of BAF complex distribution

tTo mimic primary tumor features as closely as possible, we developed organoids from primary synovial sarcoma samples removed at surgery. Tumor organoids have consistently shown superiority to cell lines in providing adequate models for the study of tumor biology (Drost & Clevers, 2018) but have not previously been established from synovial sarcoma. We generated organoids from four primary SyS (SyS1-4) and expanded them in serum-free conditions to avoid any serum-related epigenetic alteration (Dangles-Marie et al, 2007) (Fig 1A). Expression of SS18-SSX in all four organoids was confirmed by qRT-PCR (Fig 1B) and by chromatin analyses using the H3K36me3 histone mark in the SSX1 3′ terminal region as an indicator of transcription (Fig 1C). To determine SyS-specificity of the observed chromatin patterns, we used two sets of controls: two primary Ewing sarcoma organoids, established under conditions identical to those used to generate SyS organoids (Suva et al, 2009), and three different non-transformed human cell types, including primary mesenchymal stem/stromal cells (MSCs), embryonic lung fibroblasts (MRC5) and mesothelial cells (MET5A). We determined the epigenomic profiles of the primary organoids and non-transformed cells by genome-wide ChIP-seq analysis of the histone marks H3K4me1, H3K4me3, H3K9ac, H3K27ac, H3K27me3, and H3K36me3. Because SS18-SSX becomes integrated into the BAF complex (Kadoch & Crabtree, 2013), we also assessed BAF-binding sites using antibodies against the core ATPase BAF members SMARCA2/4.

ChIP-seq analysis revealed BAF ATPases to be located within domains whose breadth was far greater in SyS organoids than in any of the control cells (median width: 2,198 versus 628 bp) (Fig 1D). The observed difference in domain size was even more striking when the broadest 25% of BAF domains from each cell type were compared (Q4, Figs 1E and S1A), with a median width in excess of 5 kb in SyS cells but below 2 kb in all the other cells, including those from Ewing sarcoma-derived organoids. These differences were less striking among the narrowest 25% of BAF domains (Q1, Figs 1F and S1B) and SyS organoids contained fewer BAF sites than control cell types (Figs 1G and H and S1C). Consistent with the ability of SS18-SSX to create an aberrant BAF complex, as observed in established synovial sarcoma cell lines (Kadoch & Crabtree, 2013), our observations suggest that tumor-specific genome-wide BAF distribution is a major feature of the epigenetic landscape in SyS organoids.

## Broad BAF complex domains are associated with active chromatin states in synovial sarcoma organoids

To identify the chromatin states that are specifically associated with BAF in SyS, we focused on SMARCA2/4 sites shared among our

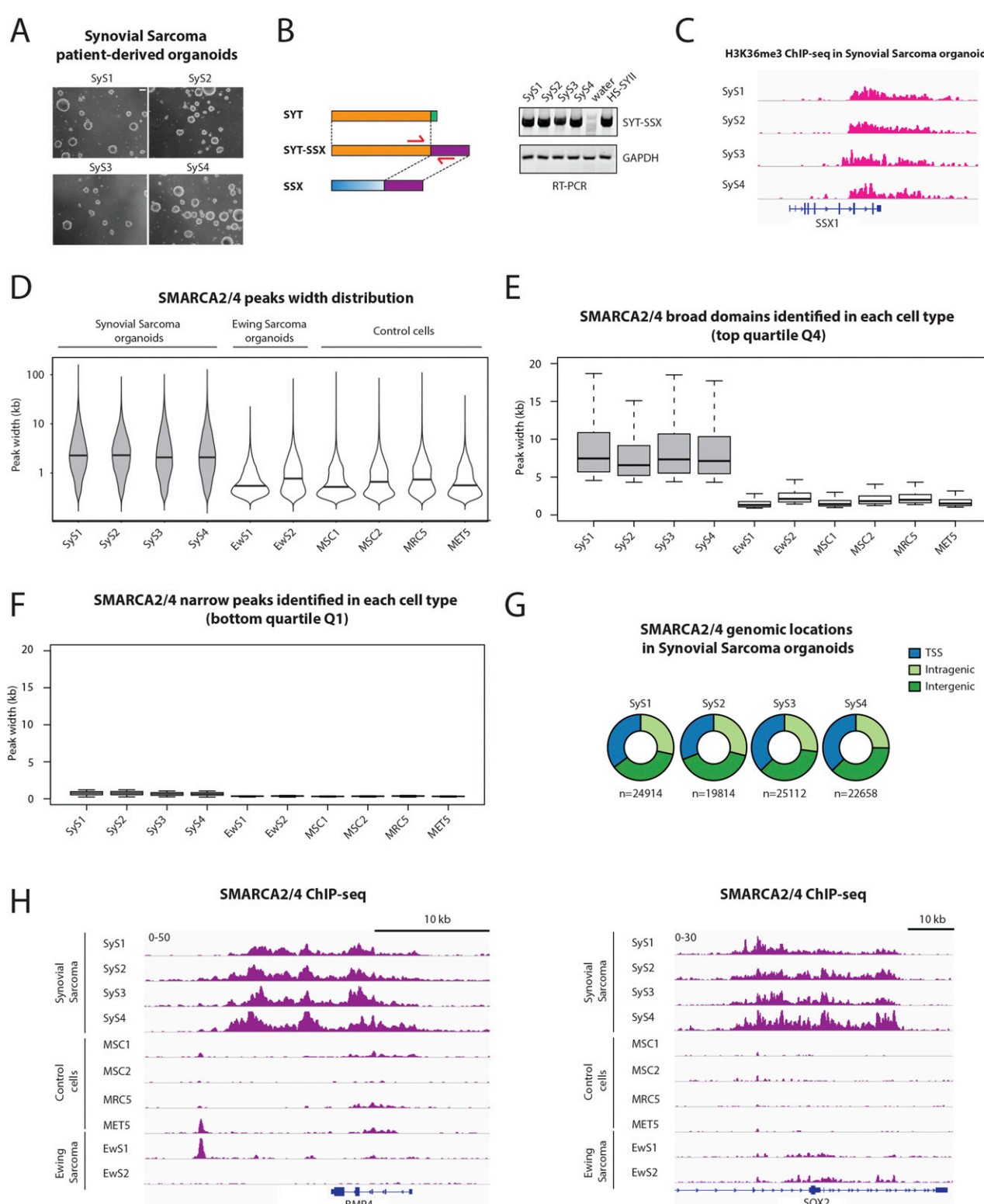

**Figure 1.  BAF complex is organized in unusually broad domains in primary synovial sarcoma organoids.**
**(A)** Micrographs show the four patient-derived synovial sarcoma 3D organoid cultures. Scale bar: 200 μm. **(B)** Schematic representation of the SS18-SSX fusion gene (left) and RT-PCR products showing the detection of SS18-SSX in patient-derived synovial sarcoma organoids. Water is used as a negative control and HSSYII cell line as a positive control. **(C)** H3K36me3 ChIP-seq signals at the SSX1 locus show active transcription of the 3′ terminal region that becomes fused to SS18 to form the fusion gene SS18-SSX in patient-derived synovial sarcoma organoids. **(D)** Violin plots show the overall distribution of peak widths for SMARCA2/4–binding sites in four synovial sarcoma patient-derived tumor organoids, two Ewing sarcoma patient-derived tumor organoids (EwS1 and 2) and four control cell types. **(E, F)** Boxplots show the

four SyS organoid models. Based on the marked differences in domain breadth between SyS and control cells (Fig 1), we first analyzed the broadest BAF domain quartile (Fig 2A and B, Q4, 3,430 peaks). These domains had a median size above 5 kb in all organoids (Figs 1E and 2A) and comparable genomic distribution (Fig S1A left and Fig 2B). We then integrated the genome-wide profiles, obtained by ChIP-seq, of our panel of histone modifications to generate a functional chromatin map in SyS organoids. Consistent with previous observations in SyS cell lines (Kadoch et al, 2017), our analysis revealed mutually exclusive distribution of broad BAF domains and the polycomb-dependent H3K27me3 mark (Fig 2C). BAF complexes were almost exclusively associated with active chromatin states as indicated by the presence of H3K4me1, H3K4me3, H3K9ac, and H3K27ac histone marks (Fig 2C and D). These observations held true at both proximal and distal sites (Fig 2D), and upon independent analysis of H3K27me3 PRC2 domains (Figs 2C–F and S2A and B, compare ChIP-seq levels between BAF domains [Fig 2D] and H3K27me3 domains [Fig S2B]). The same analyses of the narrowest 25% of BAF domains in SyS (Q1) showed increased localization to proximal sites (Fig S2C) but a similar association with active chromatin regions (Fig S2C and D). Thus, both the distinctive broad BAF domains and the more typical narrow peaks are associated with active chromatin devoid of PRC2 activity in SyS organoids.

## Broad BAF domains are associated with the expression of a tumor-specific gene signature in synovial sarcoma

To investigate the functional effect of altered BAF domains, we performed gene expression profiling of the four SyS organoids and of primary human MSCs, a candidate SyS cell of origin (Cironi et al, 2009; Garcia et al, 2012). We first compared the genes associated with either broad (Q4) or narrow (Q1) BAF domains identified in SyS (Fig 2A) and found no significant differences in relative expression levels (Fig S3A and B). We next interrogated possible Q4 BAF domain association with a distinctive gene expression signature in SyS by comparing the gene expression profiles of our SyS organoids and primary MSC cultures. Using a fourfold expression change cutoff, we identified 1,678 and 1,149 transcripts displaying, respectively, higher and lower expression in SyS compared to MSCs (Figs 3A and S3C). Remarkably, the genes that were preferentially expressed in SyS organoids tended to be linked to broad BAF domains compared to those that were more highly expressed in MSCs (median 3.27 versus 1.90 kb, $P$-value $2.2 \times 10^{-16}$) (Fig 3B). The same held true regarding the percentage of genes associated with broad (Q4) and narrow (Q1) BAF domains: a significantly higher percentage of genes that were more strongly expressed in SyS was associated with Q4 than with Q1 domains (Fig 3C, left), whereas no difference in domain association was detected among the genes that were more highly expressed in MSCs in either SyS (Fig 3C right) or in MSCs themselves (Fig S3D).

We then applied the same strategy to a set of genes that were differentially expressed between primary SyS and a panel of six unrelated sarcoma types profiled by the The Cancer Genome Atlas (TCGA) consortium (Cancer Genome Atlas Research Network. Electronic

address edsc, Cancer Genome Atlas Research Network, 2017). We first identified transcripts that are preferentially expressed in primary SyS compared to all the other sarcomas (fivefold expression cutoff, Figs 3D and S3E), and then applied the resulting signature to interrogate the corresponding BAF domain size in our organoids. Once again, we observed the genes preferentially expressed in SyS to be associated with broader BAF domains, whereas no such difference was detected among the transcripts more highly expressed in unrelated sarcomas (Figs 3E and S3F). The same enrichment in broad (Q4) BAF domains was observed (Fig 3F) among individual SyS-high gene signatures compared to those of each of the unrelated sarcomas included in the TCGA dataset (fourfold expression change cutoff Fig S3G). Taken together and consistent with observations in SyS cell lines (Kadoch et al, 2017), these observations suggest a prominent and specific role for broad BAF domains in establishing and maintaining the oncogenic gene expression signature that defines SyS cell identity.

## SS18-SSX expression leads to reversible retargeting of the BAF complex, removing the H3K27me3 repressive mark

The SS18-SSX protein has been shown to evict and replace both wt SS18 and SNF5 in BAF, creating an altered nucleosome remodeling complex that displays neomorphic oncogenic properties (Kadoch & Crabtree, 2013). We therefore reasoned that the distinctive BAF domains observed in SyS organoids may arise as a direct result of SS18-SSX expression. To date the chromatin remodeling events induced by SS18-SSX have been primarily addressed by depleting the fusion protein from SyS cell lines. Such an approach, however, does not allow assessment of the initial effect of the fusion protein in a naïve, permissive cellular context. To gain insight into the early events that drive SyS pathogenesis, we stably expressed V5-tagged *SS18-SSX1* in primary mouse C3H10T1/2 cells, which we have previously shown to provide a permissive and biologically relevant cellular model for the expression of the fusion gene (Cironi et al, 2016). C3H10T1/2 cells stably expressing SS18-SSX1-V5 and their corresponding control counterparts were profiled by ChIP-seq using the same anti-SMARCA2/4 and anti-histone mark antibodies that we applied to our SyS organoid models. Genomic distribution of the fusion protein was determined using a specific anti-V5 antibody.

Our initial analysis revealed SS18-SSX1 distribution to be reminiscent of that of the broadest Q4 BAF domains in SyS organoids. First, the fusion protein–binding sites identified in C3H10T1/2-[SS18-SSX1] cells displayed an unusual breadth (with a mean of ~9 kb, Fig 4A). Second, the genomic distribution of SS18-SSX1 recapitulated that of the broadest Q4 BAF domains in SyS organoids, with a significant enrichment at distal regulatory elements (Figs 4B and S4A). Third, consistent with its role as a core member of the BAF complex, marked increases in SMARCA2/4 ChIP-seq signals were observed at SS18-SSX1–binding sites (Fig 4C). Importantly, our analysis of genome-wide BAF domain distribution revealed increased domain

---

distribution of peak widths of the (E) broadest BAF complex domains (Q4) and the (F) narrowest BAF complex domains (Q1) identified in each cell type separately. **(G)** Pie charts show the genomic locations of SMARCA2/4–binding sites in four synovial sarcoma patient-derived tumor organoids. **(H)** Examples of broad BAF complex domains identified in synovial sarcoma at loci associated with *BMP4* and *SOX2*. **See** also Fig S1.

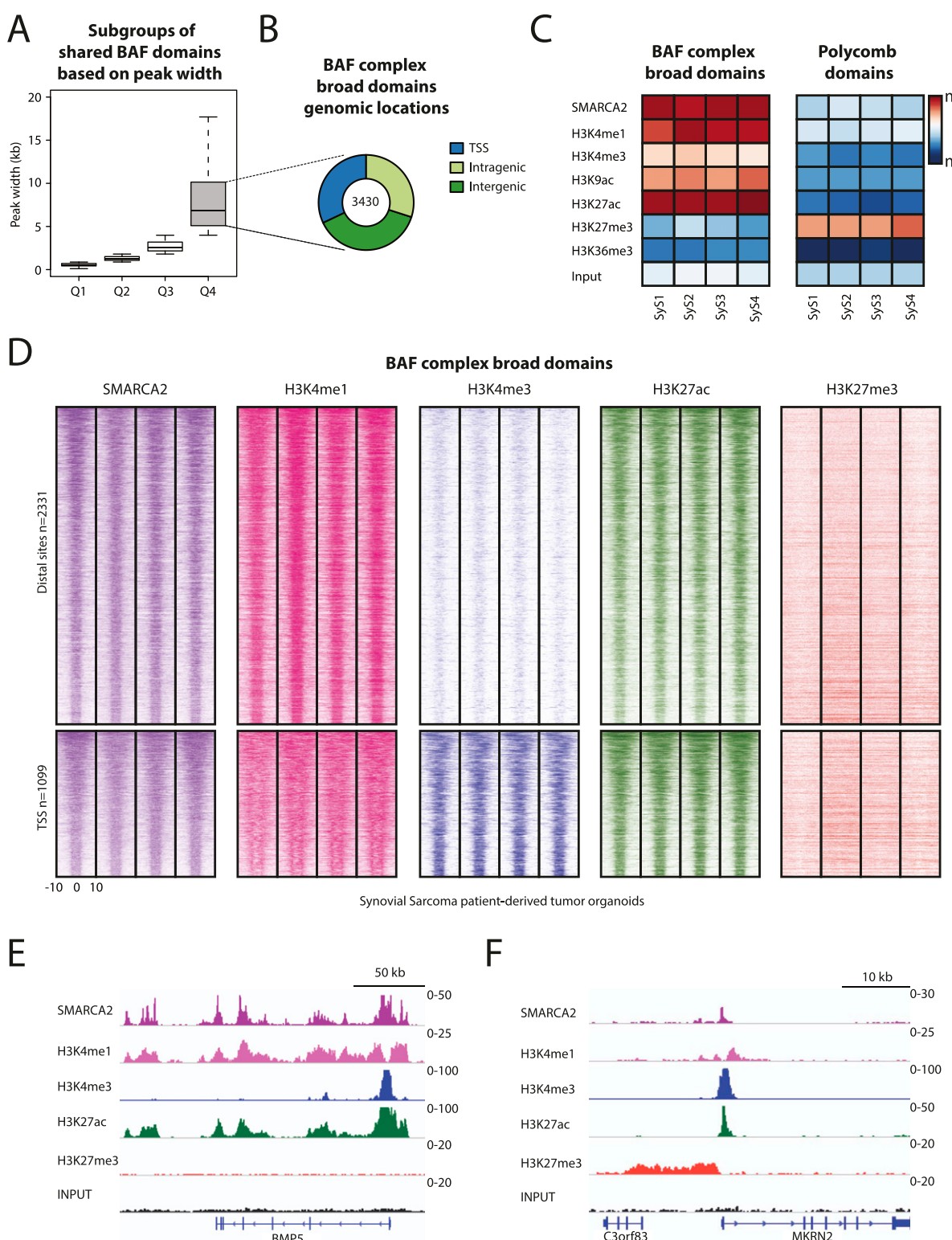

**Figure 2. BAF complex broad domains are associated with active chromatin states in synovial sarcoma.**
**(A)** Boxplot showing the distribution of peak widths for shared BAF complex domains in synovial sarcoma per quartile. **(B)** Pie chart showing genomic locations of the broadest BAF complex domains shared among synovial sarcoma tumor organoids. **(C)** Heat maps showing average ChIP-seq signals for SMARCA2/4 and the indicated histone modifications at broad BAF complex domains (*left*) and polycomb H3K27me3 domains (*right*) in synovial sarcoma. **(D)** Heat maps showing ChIP-seq signals for SMARCA2/4, H3K4me1, H3K4me3, H3K27ac, and H3K27me3 at distal sites (*top*) and promoters (*bottom*) for broad BAF complex–binding sites in synovial sarcoma. Marks of activity (H3K4me1, H3K4me3, and H3K27ac) are detected but not the polycomb repressive mark H3K27me3. 20-kb windows centered on SMARCA2/4–binding sites are

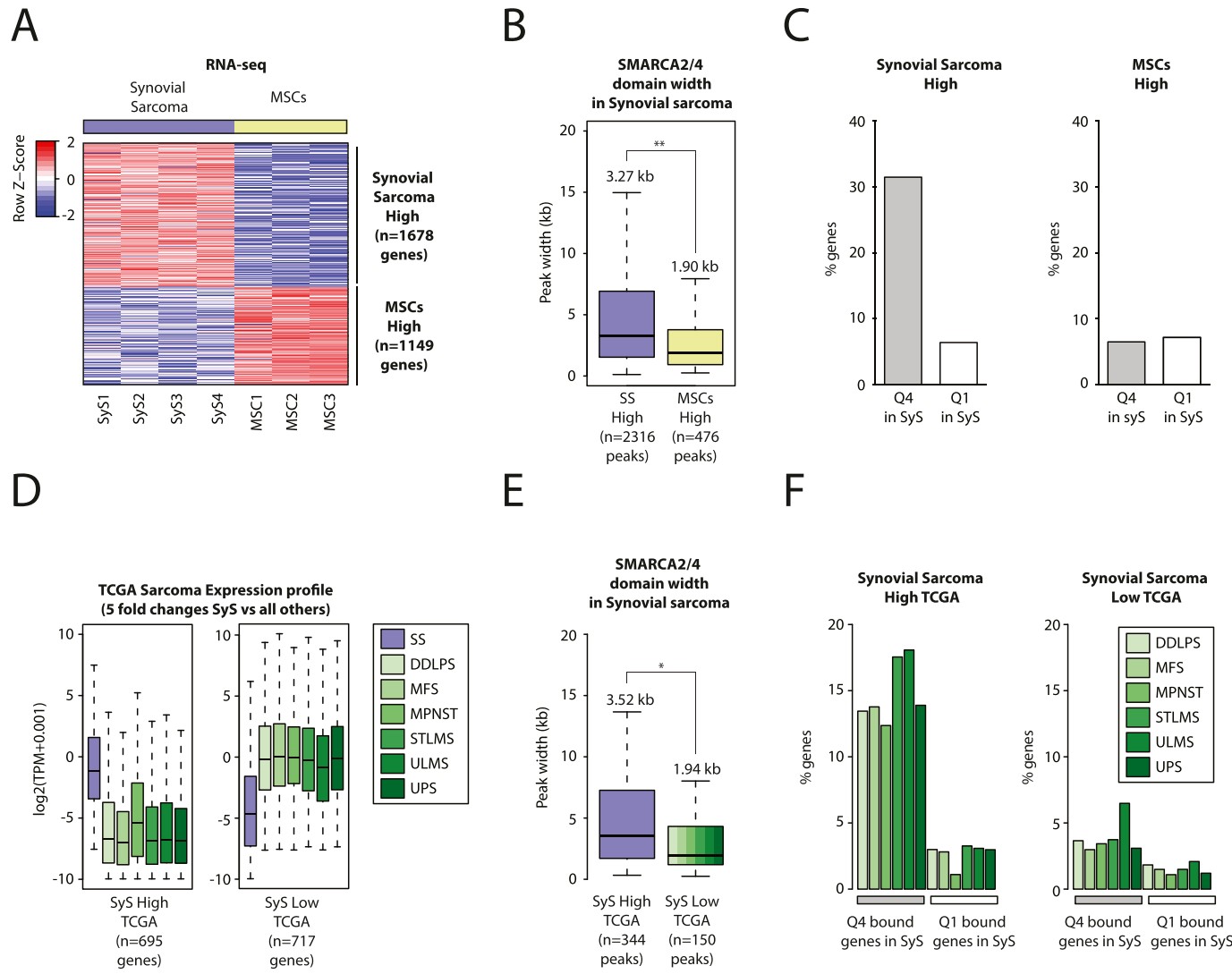

**Figure 3. Broad BAF complex domains are enriched at genes specifically expressed in synovial sarcoma.**
**(A)** Heat map showing the genes that are differentially expressed between synovial sarcoma organoids and mesenchymal stem cells (fourfold differential expression).
**(A, B)** Boxplot showing the distribution of peak widths for BAF complex domains in synovial sarcoma 3D cultures at genes highly expressed in synovial sarcoma (purple) and at genes highly expressed in mesenchymal stem cells (yellow) identified in (A). Median peak width values are indicated. **(A, C)** Barplot showing the percentage of genes identified in (A) bound by either a broad (Q4) or a narrow BAF domain (Q1) in synovial sarcoma organoids. **(D)** Boxplot showing the genes that are differentially expressed between primary synovial sarcomas and other sarcomas from The Cancer Genome Atlas (fivefold differential expression). **(D, E)** Boxplot showing the peak width distribution of BAF complex domains in synovial sarcoma organoids at genes highly expressed in synovial sarcoma (purple) versus the genes highly expressed in other sarcomas (green shades) identified in (D). Median values are indicated. **(F)** Barplot showing the percentage of genes identified by comparing primary synovial sarcomas to other sarcomas from The Cancer Genome Atlas independently (20-fold differential expression) bound by either a broad (Q4) or a narrow BAF domain (Q1) in synovial sarcoma organoids. * indicates *P*-value < 0.05, ** indicates *P*-value < 0.01 for a Welch two-sample *t* test with a 95% confidence interval. See also Fig S3.

width in C3H10T1/2-SS18-SSX1 cells (Q4, median width 4,827 versus 1,682 bp in, respectively, SS18-SSX1–expressing versus control cells, *P*-value 2.2 × 10⁻¹⁶) (Fig 4D). These remodeling events were also associated with a decrease in the overall number of BAF sites (Fig S4B), recapitulating our observations in primary SyS organoids. Interestingly, although the fusion protein could increase the width of pre-existing BAF domains, most broad BAF domains were constituted de novo upon SS18-SSX induction (Fig S4C). Taken together, these results support the possibility that reorganization and retargeting of the BAF complex in SyS is a direct effect of SS18-SSX.

Next, to evaluate the functional effect of altering the BAF complex in a naïve cellular context, we assessed the original chromatin states and genomic distribution of the 4,877 sites to which SS18-SSX is targeted in C3H10T1/2 control cells before the expression of the fusion

shown. **(E)** Representative example of BAF complex broad domain and associated histone modifications at the locus associated with BMP5 in synovial sarcoma tumor 3D cultures. **(F)** Representative example of a polycomb domain in synovial sarcoma organoids. Strong signals are detected for the polycomb repressive mark H3K27me3 but not for SMARCA2/4. See also Fig S2.

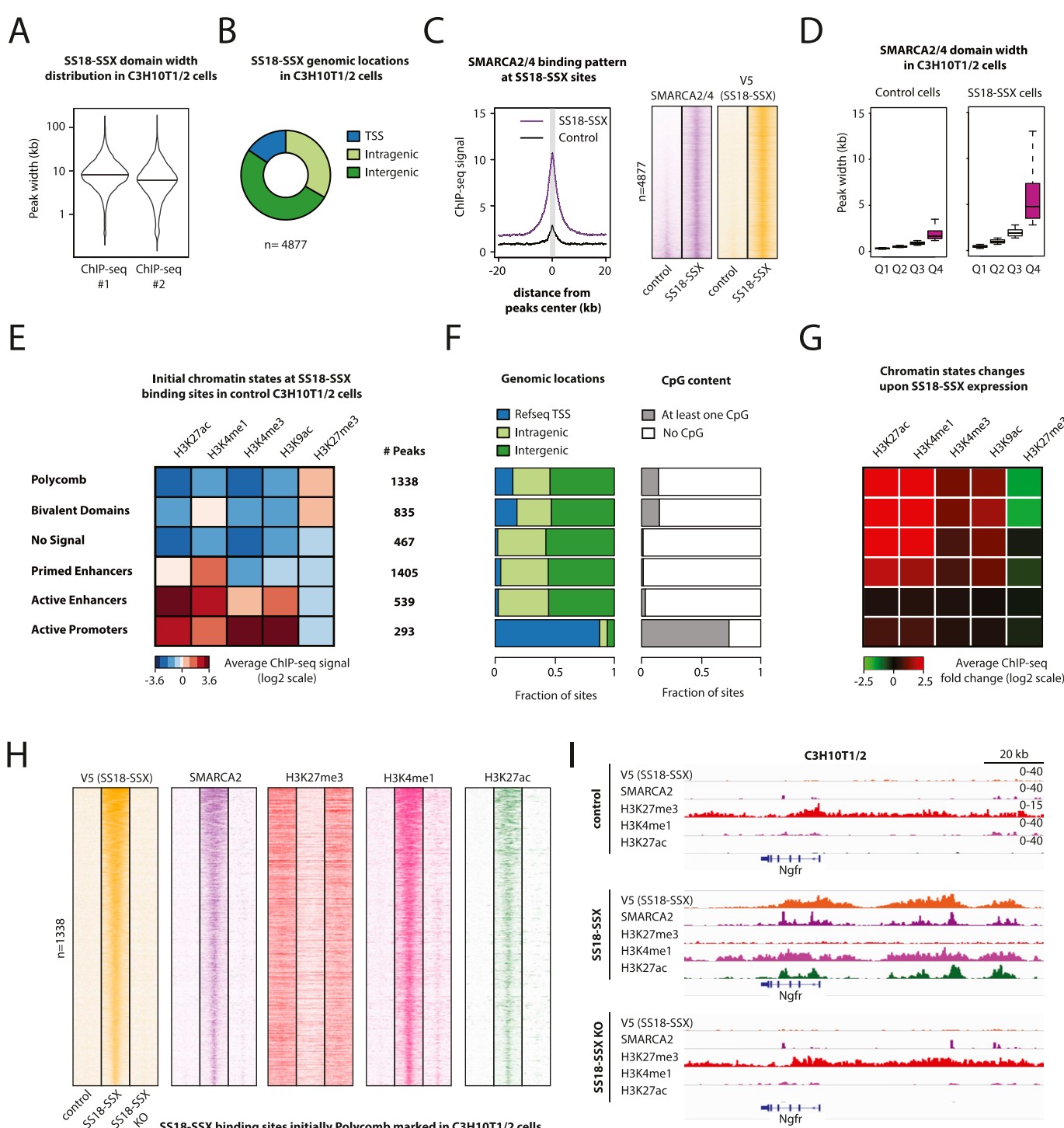

**Figure 4. SS18-SSX can induce broad BAF complex domains and evict polycomb activity.**
**(A)** Violin plot showing the overall distribution of peak widths for SS18-SSX–binding sites in C3H10T1/2 cells stably expressing the fusion protein in two independent ChIP-seq experiments. **(B)** Pie chart showing genomic locations of common SS18-SSX–binding sites in C3H10T1/2 cells stably expressing the fusion protein. **(C)** *Left*: Composite plot showing the average SMARCA2/4 ChIP-seq signals at SS18-SSX–binding sites in control cells (black) and cells expressing SS18-SSX (purple). *Right*: Heat maps showing ChIP-seq signals for SMARCA2/4 and V5 (SS18-SSX) at SS18-SSX–binding sites in control and SS18-SSX–expressing C3H10T1/2 cells. **(D)** Boxplot showing the distribution of peak widths for SMARCA2/4 in control (*left*) and SS18-SSX–expressing cells (*right*) per quartile. **(E)** Heat map showing average initial ChIP-seq signals for the indicated histone modifications in control cells at SS18-SSX–binding sites. The six indicated categories of binding sites were obtained using hierarchical clustering and manually annotated based on chromatin states. **(F)** Bar charts showing genomic locations (*left*) and annotation of CpG content (*right*) for each category of SS18-SSX–binding sites. **(E, G)** Heat map showing changes of the indicated histone modifications at each category of SS18-SSX–binding sites identified in (E). Average log$_2$ fold changes in ChIP-seq signals are displayed. **(E, H)** Heat maps showing ChIP-seq signals for V5 (SS18-SSX), SMARCA2/4, H3K27me3, H3K4me1, and H3K27ac at SS18-SSX–binding sites that bore the polycomb mark H3K27me3-in control C3H10T1/2 shown in (E) before SS18-SSX expression. 20-kb windows centered on SS18-SSX–binding sites are shown. **(I)** Representative example of broad BAF complex domains replacing preexisting H3K27me3 polycomb domains. See also Fig S4.

protein. We found that SS18-SSX is preferentially recruited to two distinct chromatin states: those bearing the polycomb mark H3K27me3 (H3K27me3 only: 1,338 sites, 27.4%; bivalent domains: 835 sites, 17.1%), and primed enhancers, marked by a high H3K4me1 to H3K27ac ratio (1,405 sites, 28.8%) (Fig 4E). It must be noted, however, that the sites marked by H3K27me3 to which SS18-SSX is recruited represent 6.54% of all the sites bearing H3K27me3. A lower proportion of SS18-SSX1–targeted sites displayed chromatin features of active promoters (293), active enhancers (539), and no signal-desert regions (467), underscoring the unique ability of the fusion protein to bind to a wide range of established active, poised or repressive chromatin states (Figs 4E and S4D). The relatively high proportion of SS18-SSX–binding sites pre-marked with H3K27me3 prompted us to investigate whether H3K27me3 itself might favor the recruitment of the fusion protein. To this end, we treated C3H10T1/2 cells with the selective EZH2 inhibitor EPZ6438 for 8 d prior to SS18-SSX expression. qPCR transcript analysis of a panel of SS18-SSX target genes, whose distal regulatory elements are marked by H3K27me3 in wild type C3H10T1/2 cells, failed to reveal differences between control and EPZ6438-treated cells upon SS18-SSX expression, despite robust reduction of H3K27me3 marks in treated cells (Fig S4E and F).

Because SS18-SSX has been recently shown to be recruited to unmethylated DNA CpG islands through direct interaction with the chromatin regulator KDM2B (Banito et al, 2018b), we interrogated the potential genomic overlap with CpG-enriched regions of the 4877 SS18-SSX1 direct binding sites identified in C3H10T1/2 cells. As expected, 73.4% of the 293 sites displaying chromatin features of active promoters harbored at least one CpG island, whereas only a minority of polycomb-marked regions (14.2%) and distal regulatory elements (3.15%) contained CpG islands (Fig 4F). These results suggest involvement of more than a single molecular mechanism in the recruitment of the fusion protein to its DNA target sites, which, in addition, may at least in part be cell type– and chromatin state–dependent. Changes in chromatin state and activity of the different categories of SS18-SSX–binding sites were then assessed upon expression of the fusion protein. Although SS18-SSX binding induced an active chromatin environment at virtually all sites (Fig 4G), the most significant changes occurred at sites that contained the repressive H3K27me3 mark (polycomb and bivalent domains, Fig 4E). These sites underwent complete removal of H3K27me3, followed by generation of fully active distal regulatory elements displaying the enhancer marks H3K4me1 and H3K27ac (Fig 4G).

Finally, to assess the reversibility of the chromatin changes mediated by SS18-SSX, we took advantage of the *lox* sites flanking the SS18-SSX1 cDNA sequence in our lentiviral vector system to genetically knockout the fusion gene in target cells. We therefore expressed a recombinant CRE in SS18-SSX–expressing C3H10T1/2 cells and measured the ensuing epigenetic changes. The same cells infected with the corresponding empty vector provided the control. After SS18-SSX1 protein depletion (Fig S5A), we observed nearly complete reversion of the chromatin states to their initial configuration in naïve C3H10T1/2 cells (Figs 4H and I and S5B). This included the complete loss of activation marks and restoration of the repressive H3K27me3 domains at SS18-SSX–binding sites. Remarkably, SS18-SSX depletion also restored the original BAF complex width observed in naïve C3H10T1/2 cells (Fig S5C), further

highlighting the tight functional relationship between altered BAF domain deposition and SS18-SSX expression. Taken together, our results reveal how expression of the SS18-SSX fusion protein in a permissive naïve cellular context results in a cascade of epigenetic changes that include retargeting of the oncogenic BAF complex to new genomic sites, and the generation of de novo active chromatin regions that reshape the epigenetic landscape of SyS precursor cells.

## SS18-SSX target gene activation is primarily mediated by the reversible eviction of H3K27me3 repressive activity

To determine the relative contribution of SS18-SSX–dependent BAF complex redistribution to target gene activation, we performed whole-genome RNA-seq expression profiling of SS18-SSX1–expressing and control C3H10T1/2 cells. Expression of the fusion protein resulted in the robust induction of 278 and repression of 113 transcripts (Fig 5A, fourfold change cutoff and false discovery rate < 0.05). Induced genes were 2.4-fold more likely to be associated with neighboring SS18-SSX binding (Fig 5B), which was mildly enriched at proximal elements (Fig 5C). Consistent with our preceding results, we found that most fusion protein–binding sites associated with induced transcripts had a repressive chromatin configuration before SS18-SSX recruitment (polycomb-only or bivalent, Fig 5D, right). Binding of the fusion protein was followed by complete removal of the H3K27me3 mark and the concomitant deposition of the chromatin features of active proximal and distal regulatory elements (Fig 5E, right). In contrast, the chromatin states at fusion protein–binding sites associated with repressed transcripts were enriched in primed or active enhancer marks (Fig 5D, left) and did not undergo substantial changes upon SS18-SSX recruitment (Fig 5E, left). Finally, consistent with the observed chromatin state reversibility upon SS18-SSX depletion (Fig 4H and I), CRE expression in C3H10T1/2$^{SS18-SSX1}$ cells resulted in a significant reversion of their transcriptional profile to that preceding SS18-SSX introduction (Fig 5F), the lack of complete reversion being most likely due to incomplete depletion of SS18-SSX. Taken together, these results highlight the notion that PcG domain presence, although not a requisite for SS18-SSX recruitment, is linked to the most significant SS18-SSX-induced chromatin changes associated with transcriptional activation. Moreover, both chromatin and gene expression changes remained reversible upon SS18-SSX depletion, suggesting that detailed understanding of the mechanisms that underlie the observed plasticity may provide a tangible opportunity to block the oncogenic properties of the fusion protein.

## SS18-SSX orchestrates functional PRC1–PRC2 complex uncoupling

Similar to the aberrant fusion proteins that drive the development of other human sarcomas, SS18-SSX lacks an enzymatic domain amenable to direct pharmacological targeting. Accordingly, detailed understanding of the chromatin remodeling events that drive transcriptional changes at the binding sites of the fusion protein and their reversibility upon SS18-SSX depletion are critical toward identifying new tumor-specific therapeutic strategies. Our experiments so far have shown a pattern of mutual exclusivity between SS18-SSX binding and H3K27me3 deposition, suggesting genome-wide

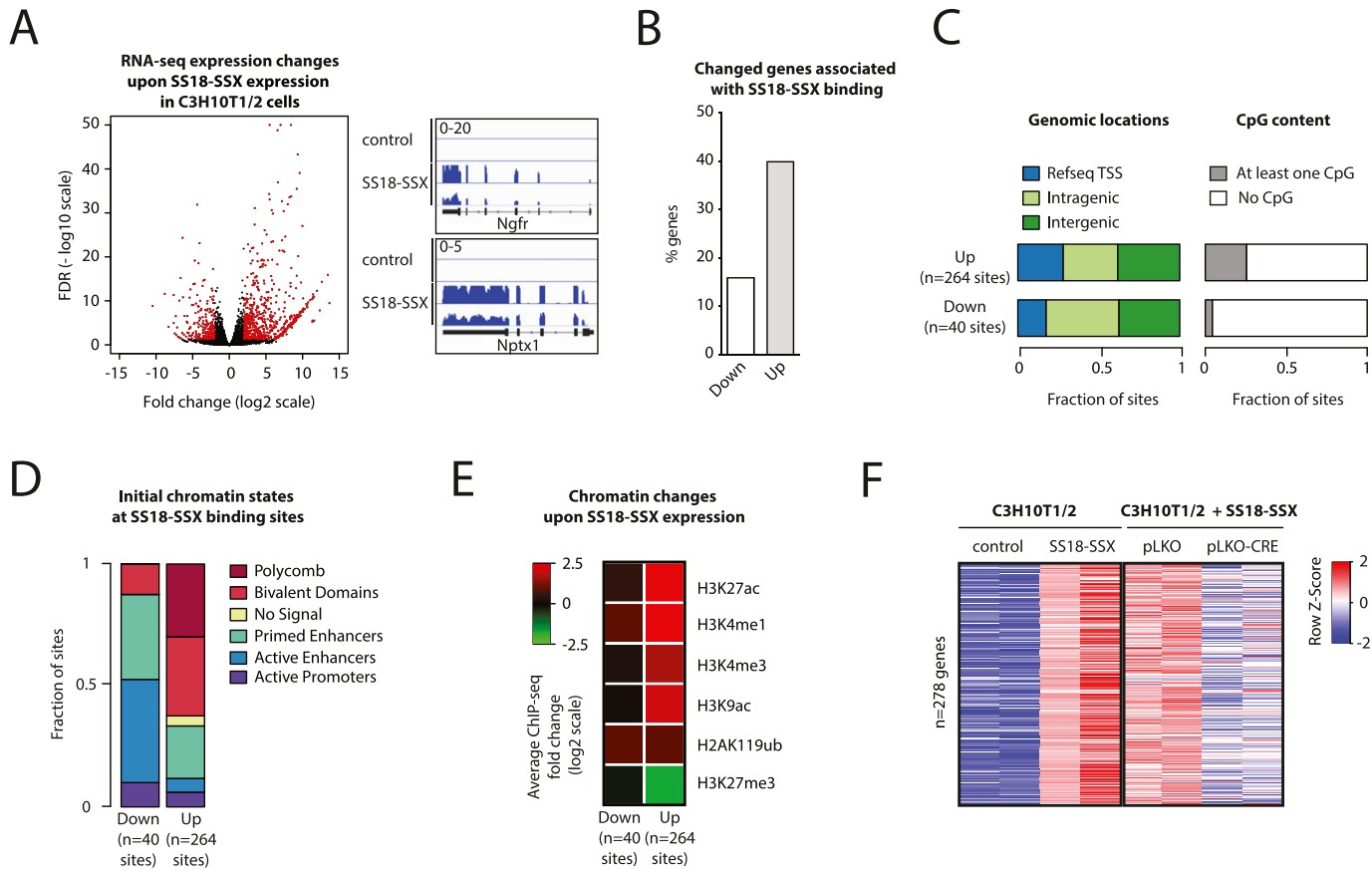

**Figure 5. SS18-SSX can induce strong gene expression changes through reversible chromatin modification mechanisms.**
**(A)** *Left*: Volcano plot showing gene expression changes upon stable SS18-SSX expression in C3H10T1/2 cells. Red dots indicate genes differentially expressed (at least fourfold change and adjusted *P*-value < 0.05). *Right*: Representative examples of genes strongly activated upon SS18-SSX expression. **(B)** Barplot showing the percentage of up- and down-regulated genes upon SS18-SSX expression associated with SS18-SSX binding. **(C)** Bar charts showing genomic locations (*left*) and annotation of CpG content (*right*) of SS18-SSX–binding sites associated with up- and down-regulated genes. **(D)** Bar chart showing the distribution of initial chromatin states as defined in Fig 4E at SS18-SSX–binding sites associated with up- and down-regulated genes. **(E)** Heat map showing ChIP-seq changes for the indicated histone modifications at SS18-SSX–binding sites associated with up- and down-regulated genes. **(F)** Heat map showing gene expression changes upon SS18-SSX expression in C3H10T1/2 cells (*left*) and upon SS18-SSX removal (*right*). See also Figs S5 and S6.

functional antagonism between PRC2 and oncogenic BAF complexes. Consistent with this notion, binding of the BAF complex has been shown to result in the rapid eviction of both canonical PCR1 and PRC2 complexes in mouse embryonic fibroblasts through the ATPase activity of SMARCA4 (Stanton et al, 2017). More recently, SS18-SSX itself has been found to recruit ncPRC1.1 at its binding sites in SyS cells (Banito et al, 2018b). We reasoned that a possible explanation for these observations may be linked to the functional uncoupling of the PRC1–PRC2 complexes at the fusion protein–binding sites in SyS, starting with eviction of PRC2 followed by the recruitment of ncPRC1.1 and transcriptional activation. An analogous uncoupling event has been previously shown to induce gene expression in models of epidermal and leukemia stem cells (van den Boom et al, 2016; Cohen et al, 2019).

To test this hypothesis, we used ChIP-seq to profile the PRC1 core member RING1B and its related H2AK119ub histone mark in control and SS18-SSX–expressing C3H10T1/2 cells. We initially focused on the 1338 PRC2-marked genomic regions that undergo H3K27me3 removal upon SS18-SSX expression and compared the signal levels of RING1B, H2AK119ub, and H3K27me3 between control and SS18-SSX–expressing cells. Remarkably, binding of the fusion protein increased the signal of both RING1B and the H2AK119ub mark at these genomic regions, concomitant to nearly complete removal of H3K27me3 (Fig 6A and B). Importantly, SS18-SSX recruited ncPRC1 to all of its binding sites and not exclusively to those bearing the H3K27me3 mark (Fig S6). However, neither RING1B nor H2AK119ub global protein levels were increased in SS18-SSX–expressing cells (Fig S7A). Consistent with our previous results, we found these chromatin changes to be reversible upon CRE-mediated depletion of the fusion gene in C3H10T1/2 cells, underscoring their plasticity (Figs 6B and S7B). We also identified strong RING1B and H2AK119ub ChIP-seq signals at the broad Q4 SMARCA2/4 domains in SyS organoids at similar or even higher levels than H3K27me3 marked polycomb sites (Fig 6C and D). Furthermore, similar to our initial observations on BAF complex binding patterns (Fig 4C and D), SS18-SSX augmented RING1B domain width upon expression in C3H10T1/2 cells (Fig 6E, median width 1,229 versus 756 bp in, respectively, SS18-SSX1–expressing versus control cells, *P*-value 2.2 × 10⁻¹⁶), and in both experimental models, the broadest RING1B domains were the most enriched at SS18-SSX bound genomic locations in C3H10T1/2 cells

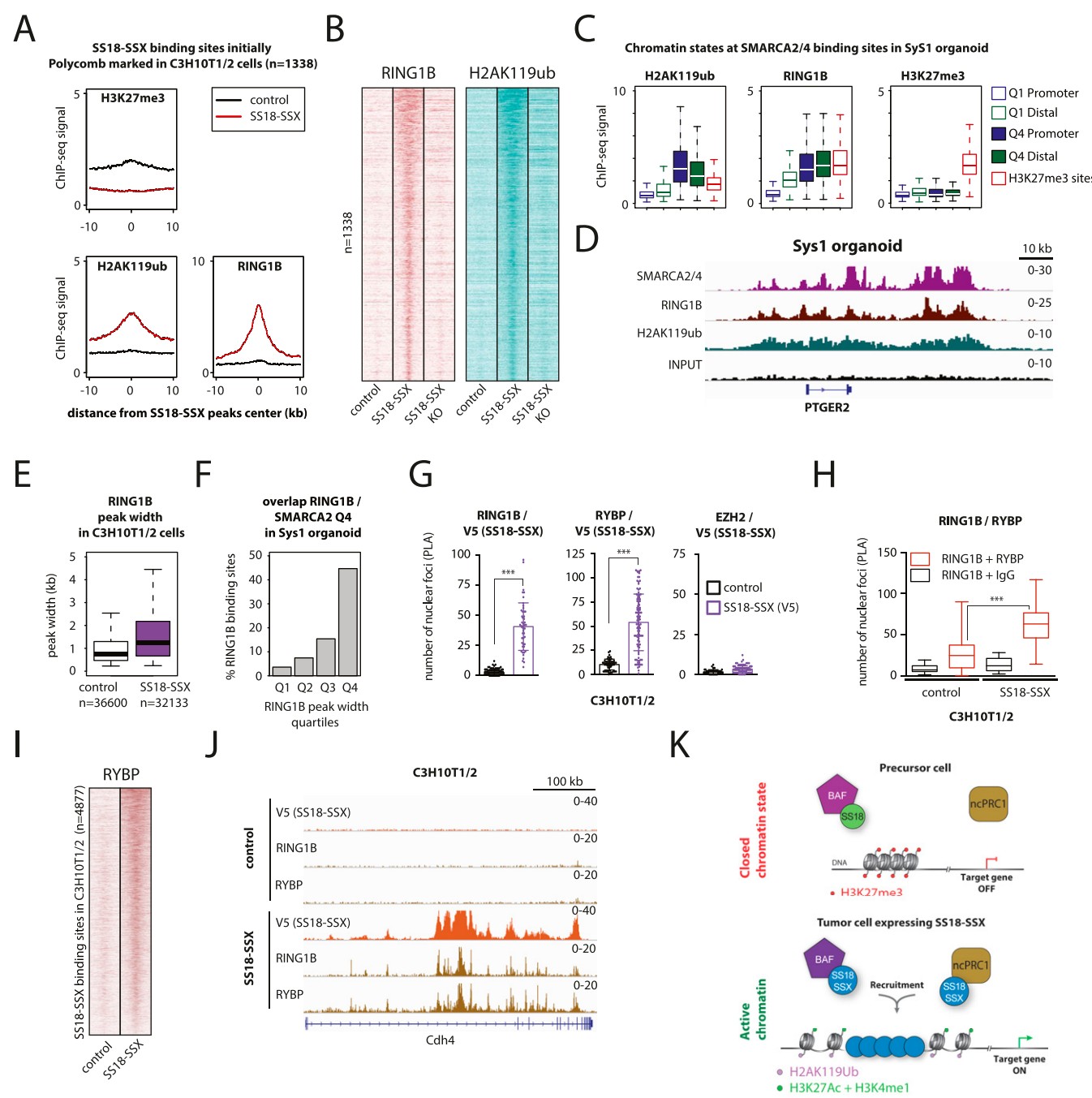

**Figure 6. SS18-SSX expression results in a functional PRC1–PRC2 complex uncoupling at its genomic binding sites.**
**(A)** Composite plots for RING1B, H2AK119ub, and H3K27me3 signals at 1338 SS18-SSX–binding sites, initially bearing the PRC2 repressive mark, showing that expression of the fusion protein in C3H10T1/2 cells results in an increase in the RING1B and H2AK119ub signals, and the removal of the H3K27me3 mark. **(A, B)** Heat maps for RING1B and H2AK119ub signals at the same 1338 SS18-SSX–binding sites as in (A), illustrating the reversibility of the chromatin remodeling pattern upon CRE-mediated SS18-SSX depletion in C3H10T1/2 cells. 20-kb windows centered on SS18-SSX–binding sites are shown. **(C)** Boxplot analysis of the RING1B, H2AK119ub, and H3K27me3 ChIP-seq signals in primary SyS 1 organoids, confirming the presence of PRC1 and the related H2AK119ub mark, but not the PRC2 mark H3K27me3, at broad BAF domains in primary SyS tumor models. **(D)** Representative example of RING1B recruitment and H2AK119ub deposition at a broad BAF complex domain in SyS organoid cultures. **(E)** Boxplot showing the distribution of peak widths for RING1B in control and SS18-SSX–expressing C3H10T1/2 cells. **(F)** Barplot depicting the percentage of RING1B-binding sites in each width quartile that overlap with the broadest SMARCA2/4 domains in SyS organoids. **(G)** Proximity ligation assay analyses demonstrate direct interactions between SS18-SSX and the PRC1 subunits RING1B and RYBP, but not the PRC2 core member EZH2, in C3H10T1/2 cells. **(H)** Proximity ligation assay analysis of SS18-SSX–expressing or control C3H10T1/2 cells confirming that the expression of the fusion protein enhances the assembly of the ncPRC1, as indicated by the increase in interactions between its core members RING1B and RYBP. **(I)** Heat map showing ChIP-seq signals for RYBP at SS18-SSX–binding sites in C3H10T1/2 cells. 20-kb windows centered on SS18-SSX–binding sites are shown. **(J)** Representative example of RING1B and RYBP recruitment at SS18-SSX–binding sites in C3H10T1/2 cells. **(K)** A mechanistic model of SS18-SSX chromatin remodeling activity in SyS. After SS18-SSX expression, the PRC2 repressive mark H3K27me3 is replaced by the active histone modifications H3K4me1 and H3K27ac. The concomitant recruitment of a non-canonical PRC1 by the fusion protein leads to PRC2-PRC1 uncoupling, H2AK119ub deposition and target gene activation. *** indicates *P*-value < 0.0001. Statistical analyses were performed by *t* test. See also Fig S7.

(Fig S7C) and at the broad Q4 SMARCA2/4 domains in SyS organoids (Figs 6F and S7D).

We then used proximity ligation assay (PLA) to validate the interaction between the fusion protein and the PRC1 complex members RING1B and RYBP on the one hand and the absence of interaction between SS18-SSX and the PRC2 core member EZH2 on the other (Fig 6G). Whereas RING1B is a core member of all PRC1 complex variants, RYBP is primarily associated with ncPRC1 complexes. We therefore reasoned that by recruiting de novo PRC1 variants at its binding sites, SS18-SSX might increase detectable interactions between RING1B and RYBP. To test this hypothesis, we analyzed the PLA interaction signal between RING1B and RYBP in control and SS18-SSX–expressing C3H10T1/2 cells, and identified a marked increase in the RING1B–RYBP interactions in the presence of the fusion protein (Fig 6H), which was not observed between the PRC2 core members EED and EZH2 (Fig S6E). Consistent with these observations, RYBP signals detected by ChIP-seq were also strongly increased at SS18-SSX–binding sites in C3H10T1/2 cells (Fig 6I and J). In contrast to RING1B, cellular expression of RYBP was increased by SS18-SSX, which may reflect its implication in facilitating the transition of canonical PRC1 to ncPRC1 at the fusion protein–binding sites (Fig S7A). Taken together, these results demonstrate that SS18-SSX induces functional uncoupling of the canonical PRC2-PRC1 complexes at its direct binding sites, by recruiting a ncPRC1 variant that sustains transcriptional activation (Fig 6K).

### USP7 depletion is an epigenetic vulnerability in synovial sarcoma

Because reconfiguration of chromatin remodeling complexes may provide new opportunities to uncover epigenetic vulnerabilities, we interrogated the DepMap database of cancer dependencies for SyS-selective vulnerability to inactivation of any of the six PRC1 variants (Vidal & Starowicz, 2017). Comparison of RNAi-induced cell vulnerabilities between SyS cell lines and all other cell lines revealed that among the 21 genes analyzed, *USP7* depletion produced a consistent and robust detrimental effect that was significantly more pronounced in SyS (Fig 7A). Ubiquitin-specific protease 7 (USP7) is a member of the deubiquitinase (DUBs) family and has an important implication in cancer development by altering the DNA damage response, apoptosis and cell cycle control (Nicholson and Suresh Kumar, 2011; Smits & Freire, 2016). Functionally, USP7 participates in stabilizing ncPRC1.1, in which it acts as a regulator of H2AK119ub deposition and gene expression (Wheaton et al, 2017). To test whether USP7 may represent a druggable epigenetic vulnerability of SyS directly linked to SS18-SSX, we tested the genome-wide localization of USP7 by ChIP-seq and found strong signals at SS18-SSX–binding sites in C3H10T1/2 cells (Fig 7B and C). Although no change in the USP7 protein level was observed upon SS18-SSX expression (Fig S8A), the fusion protein induced a remarkable reorganization of USP7-binding sites genome-wide (Fig S8B and C). Consistent with these results, we also found evidence of interaction between SS18-SSX and USP7 by PLA in C3H10T1/2 cells and the HSSYII SyS cell line that harbors an endogenous HA-tagged SS18-SSX protein (Banito et al, 2018a) (Fig 7D and E). These observations prompted us to validate the effect of *USP7* depletion in SyS cells. Using two different single guide RNAs specifically targeting the *USP7* coding sequence we achieved significant reduction

in USP7 expression in two SyS cell lines, HSSYII and SYO1 (Fig S8D). As previously observed in leukemia, *USP7* depletion in SyS resulted in the partial disassembly of the ncPRC1 complex, as illustrated by the marked decrease in interactions between RING1B and RYBP in the SyS cell line HSSYII (Fig 7F). Importantly, loss of *USP7* did not affect the expression levels of SS18-SSX in either cell line (data not shown). Consistent with its selective detrimental effect observed in DepMap, *USP7* depletion produced a significant decrease in proliferation of the SyS HSSYII and SYO1 cell lines, but not of the Ewing sarcoma A673 and RDES lines, despite comparable baseline protein expression levels (Fig 7G). Accordingly, synovial sarcoma cells (as represented by two cell lines and one organoid model) displayed markedly higher sensitivity to incremental doses of the USP7 inhibitor FT827 than Ewing sarcoma cells (Fig S8E). Taken together, these results identify an epigenetic dependency of SyS cells to USP7 depletion that is directly linked to the molecular function of SS18-SSX.

## Discussion

In the present work, we generated primary organoid models to study the mechanisms that drive SyS. Whereas cell lines have since their first establishment provided the most widely used means to study diverse cancer types, there is increasing evidence that organoids reflect key biological properties of primary tumors more accurately, including response to therapy and intra-tumor heterogeneity (Tuveson & Clevers, 2019; Schutgens & Clevers, 2020). Consistent with this notion, our work on primary Ewing sarcomas has shown that organoids can be valuable tools for elucidating mechanisms of tumor initiation and progression in sarcomas (De Vito et al, 2012; Cornaz-Buros et al, 2014). Because SyS is driven by an aberrant chromatin regulator, we used genome-wide chromatin profiling to map the epigenetic landscape of SyS organoids and define the chromatin remodeling events induced by the SS18-SSX fusion protein. Our results show that SS18-SSX establishes a distinctive SyS chromatin landscape by retargeting BAF complexes to new genomic locations, from which the repressive histone modification H3K27me3 deposited by PRC2 is removed and to which a ncPRC1 complex is recruited. Although our observations suggest that SS18-SSX recruits the ncPRC1.1 complex containing RING1B, RYBP, and USP7, further investigation will be required to elucidate the exact composition and function of this protein complex in SyS. Importantly, we find that these major chromatin changes are completely reversible upon SS18-SSX removal, suggesting that their effectors constitute potentially attractive therapeutic targets.

Our results support recent studies that showed retargeting of the oncogenic BAF complex by SS18-SSX in SyS cell lines and its antagonistic effect on the H3K27me3 repressive mark (Kadoch & Crabtree, 2013; Kadoch et al, 2017). Intriguingly, short-term recruitment experiments show that both wild type BAF and SS18-SSX–containing BAF complexes can directly and rapidly evict PRC1 and PRC2 (Kadoch & Crabtree, 2013; Kadoch et al, 2017). The recruitment of a ncPRC1 observed at later time points in our work may thus occur after the initial removal of PRC1 and PRC2. Alternatively, the recruitment of ncPRC1 may occur more readily in the chromatin

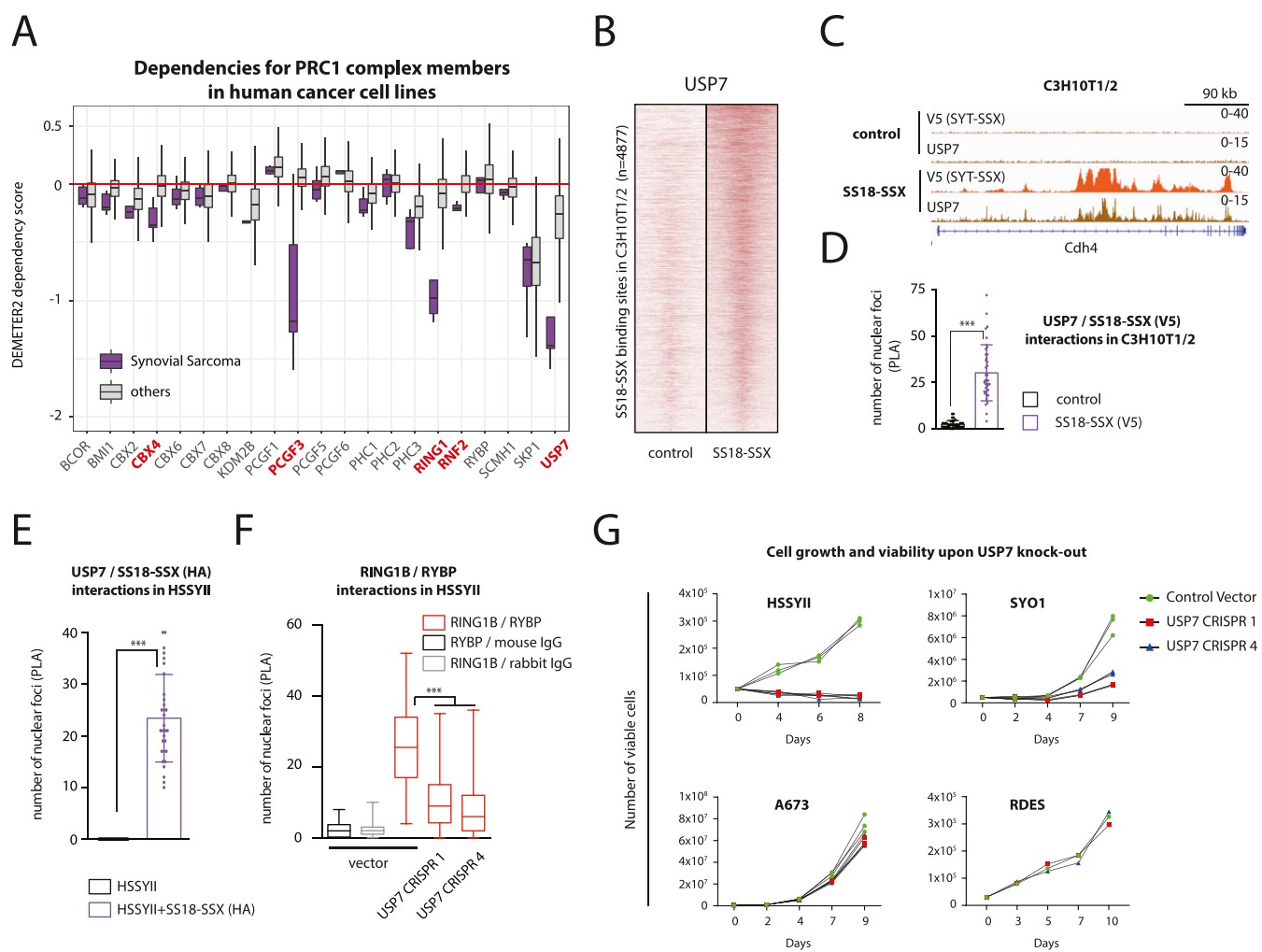

**Figure 7. USP7 depletion constitutes an epigenetic vulnerability in synovial sarcoma.**
**(A)** Gene dependency score analysis for a panel of PRC1 members in SyS versus all other cell lines present in the DepMap database identifies USP7 as a selective vulnerability in SyS. Genes shown in red had an adjusted *P*-value < 0.05. **(B)** Heat map showing ChIP-seq signals for USP7 at 4877 SS18-SSX–binding sites in C3H10T1/2 cells. 20-kb windows centered on SS18-SSX–binding sites are shown. **(C)** Representative example of USP7 recruitment by SS18-SSX at its binding sites in C3H10T1/2 cells. **(D, E)** Proximity ligation assay analysis demonstrates direct interaction between USP7 and SS18-SSX in C3H10T1/2 and HSSYII SyS cells. **(F)** Proximity ligation assay shows a significant decrease in interactions between the ncPRC1 members RING1B and RYBP following USP7 removal in HSSYII cells. **(G)** Cell viability assays in SyS (HSSYII, SYO1) and EwS (A673, RDES) cells upon CRISPR-mediated *USP7* KO confirm the selective detrimental effect of USP7 depletion in the SyS cells. *** indicates *P*-value < 0.0001. Statistical analyses were performed by *t* test. See also Fig S8.

environment specific to precursor cells such as CH310T1/2. Whereas polycomb group proteins are traditionally considered to be epigenetic repressors, recent evidence suggests a more complex scenario in which ncPRC1 plays an important physiological role associated with gene activation (Cohen et al, 2019; Wang & Yi, 2019). For example, loss of ncPRC1 in epithelial cells of the skin results in decreased expression of genes involved in cell adhesion and cytoskeletal organization (Cohen et al, 2019). Similarly, ncPRC1.1 binds a set of genes that are involved in promoting leukemogenesis, lack the H3K27me3 mark, and are associated with transcriptional activity in acute myeloid leukemia stem cells (van den Boom et al, 2016). Accordingly, the assembly of new ncPRC1 complexes at SS18-SSX–binding sites in CH310T1/2 cells may drive the expression of a gene repertoire that sustains SyS development and/or maintenance.

The factors that control the recruitment of SS18-SSX to specific genomic locations remain to be fully elucidated. Recent studies have shown that interactions between SS18-SSX and the ncPRC1.1 member KDM2B may account for the recruitment of SS18-SSX to unmethylated CpG islands (Banito et al, 2018b). However, our study shows that in addition to these locations, there is widespread recruitment of SS18-SSX to H3K27me3 enriched regions and distal regulatory sites devoid of CpG islands, suggesting involvement of additional mechanisms. Among potential candidates, transcription factors could provide a natural alternative. A possible scenario that emerges from our results is that reconfiguration and retargeting of BAF complexes by SS18-SSX creates broad accessible chromatin regions in SyS precursor cells, allowing a repertoire of transcription factors to bind previously inaccessible sites. Binding to such sites may generate new regulatory elements and gene expression programs that promote transformation and establishment of SyS identity. The final outcome may therefore depend on the repertoire

of preexisting developmental transcription factors in cells that acquire the chromosomal translocation, which may in turn determine permissiveness to the transforming potential of SS18-SSX.

In summary, our results show that the chromatin landscape of SyS organoids is shaped by chromatin regulation events downstream of the SS18-SSX fusion protein. Given that our experiments reveal these features to be fully reversible, targeting the mechanisms that establish and maintain oncogenic gene regulation in SyS is likely to lead to new therapeutic opportunities. For example, the connection between SS18-SSX and ncPRC1 illustrated by our results led us to consider the PRC1.1 core member USP7 as a potential therapeutic target. Accordingly, we find that USP7 down-regulation leads to the deconstruction of the ncPRC1 complex in SyS cells, and a marked decrease in their proliferation. SyS organoids can thus serve as a powerful model to define critical epigenetic mechanisms driven by SS18-SSX and test their potential therapeutic value.

# Materials and Methods

### MSCs and tumor organoids

Primary tumor samples and MSCs were collected with approval from the Institutional Review Board of the Center Hospitalier Universitaire Vaudois (University of Lausanne). Samples were de-identified before our analysis. Synovial sarcoma and Ewing sarcoma patient-derived tumor organoids were cultured in IMDM (Gibco), supplemented with 20% KO serum (Gibco), 10 ng/ml human recombinant EGF and basic fibroblast growth factor (Invitrogen), and 1% Pen/Strep (Gibco) in ultra-low attachment flasks (Corning), as previously described (Suva et al, 2009). The presence and type of the specific translocation was confirmed by RT-PCR and sequencing of the PCR product. Primary bone marrow-derived or tissue-derived MSCs were obtained as previously described (Riggi et al, 2008) and cultured in Iscove's modified Dulbecco's medium containing 10% FCS and 10 ng/ml platelet-derived growth factor BB (PeProTech). FACS analysis for standard surface markers and karyotype analysis were performed in each isolated population to confirm cell identity as normal MSCs.

### Cell lines

C3H10T1/2 cell line, Ewing sarcoma cell lines RDES and A673, MET5A and MRC5 cells were obtained from American Type Culture Collection and were cultured according to American Type Culture Collection recommendations. HSSY-II cells were purchased from RIKEN BioResouce Center and SYO1 (RRID:CVCL_7146) were a gift from Dr A Kawai (Okayama university graduate school of medicine, dentistry and pharmaceutical sciences, Japan). HSSYII cells bearing the HA-tagged SS18-SSX1 protein were a gift from Dr Ana Banito (DFKZ German Cancer Center). All synovial sarcoma cell lines were cultured in DMEM, 10% FBS, 1% penicillin/streptomycin.

### Lentiviral infection, protein expression, and knockdown

Lentiviruses were produced in 293T cells by transfection with gene delivery vector and packaging vectors GAG/POL and VSV plasmids using Fugene6 reagent (Promega). Viral supernatants were collected 72 h after transfection and concentrated by centrifugation. Virus-containing pellets, resuspended in DMEM, were added dropwise on cells in presence of media supplemented with 8 μg/ml polybrene. Infected cells were selected with puromycin used at 1 μg/ml.

Expression of SS18-SSX1-V5 was achieved using the self-inactivating lentiviral Gene Transfer and Expression system pLIVc which produces a floxed proviral genome (Cironi et al, 2016). Knockdown of the transgene (SS18-SSX-V5) was achieved by removing the provirus genome (flanked by two LOXp sites) through the expression of the CRE recombinase obtained by infection with the LV-Cre pLKO.1 lentiviral plasmid (25997; Addgene). CRISPR/Cas9 genome editing of cells for the knockout of USP7 was obtained by infection with the lentiCRISPR v2 plasmid containing the following specific single guide RNA: AGACACCAGTTGGCGCTCCG, TCTTCAGCACTGCTTGTGCA (Genscript). The efficiency of each overexpression or knockdown was determined by Western blot and qRT-PCR analyses.

### Western blot analysis

Western blotting was performed using standard protocols. Primary antibodies used were mouse anti-V5 (Invitrogen), rabbit anti-USP7 (A300-033A; Bethyl Laboratories), rabbit anti-RING2 (187509; Abcam), mouse anti-RING2 (sc-101109; Santacruz Biotechnology), monoclonal anti-HA (Covance Research Product Inc.), mouse anti-tubulin (Calbiochem), and mouse anti-actin clone AC-40 (Sigma-Aldrich). Secondary antibodies were goat antirabbit horseradish peroxidase-conjugated (Dako) and sheep antimouse horseradish peroxidase-conjugated (Amersham). Membranes were developed using SuperSignal west pico plus chemioluminescent substrate (Thermo Fisher Scientific) or WesternBright Sirius Detection Kit (Advansta). Bands were visualized using the Fusion FX machine from Vilbert-Loumat. Densitometric analyses were performed using ImageJ software (https://imagej.nih.gov/ij/).

### PLA

PLA was performed using a Duolink II Fluorescence PLA kit (Olink Bioscience) as instructed by the manufacturer. The cells were seeded at 70% confluence in 0.2 cm$^2$ dishes, fixed in 4% paraformaldehyde in PIPES buffer for 13 min at RT, and permeabilized with 0.3% triton in PBS for 3 min. The following primary antibodies were used at the specified dilutions: 1: 2,000 for mouse anti-V5 (Invitrogen), rabbit anti-V5 (Ab15828; Abcam), and monoclonal anti-HA (HA.11; Covance); 1:1,000 for rabbit anti RYBP (Ab185971; Abcam), rabbit anti RING2 (Ab187509; Abcam), rabbit anti EED (Ab4469; Abcam), and mouse anti EZH2 (H.547.3; Thermo Fisher Scientific); 1:500 for mouse anti RING2 (sc-101109; Santacruz Biotechnology) and 1:10,000 for goat anti USP7 (PLA0306; Sigma-Aldrich). PLA amplification was labeled with Alexa Fluor 594 (Olink Bioscience). Slides were counterstained with DAPI, mounted, and imaged using the Zeiss Confocal Fluorescent Microscope LSM710, with an oil immersion objective 63×, NA 1.4. For each channel the pin hole was set to 0.9 AU. For each sample the Z-stack was acquired with a line averaging of two passages and with a sampling in the XYZ according

to the optimal Nyquist criteria. Before analysis, the Z-stack was converted with maximum intensity projection. The resulting images were analyzed using ImageJ software (https://imagej.nih.gov/ij/) as previously described (Cironi et al, 2016). For statistical analysis, fluorescent foci were counted for each sample in five different fields each containing an average of 8–10 cells.

## Cell growth, viability assays, and EPZ6438 treatment

Global decrease in H3K27m3 was obtained by treating C3H10T1/2 cells for 8 d with 10 μM EPZ6438 (Selleckchem). The drug was refreshed every second day. At day 8 C3H10T1/2 cells were infected with V5-tagged SS18-SSX or empty pLIVc vector and harvested 72 h later to produce total cell lysates and RNA for further analyses. 5–7 d after infection and puromycin selection, HSSYII, SYO1, A673, and RDES populations showing a substantial USP7 knockdown were established. USP7 knockdown cells or control cells were quantified using an automated cell counter Countess II (Thermo Fisher Scientific) and seeded in 60-mm plates at a concentration of 500,000 cells/plate. Cell count was performed in triplicates at regular intervals over a period of time of about 10 d. Trypan blue exclusion was used to evaluate cell viability.

## ChIP-seq

ChIP assays were carried out on ~2–5 million cells per sample and per epitope, following the procedures described previously (Mikkelsen et al, 2007). In brief, chromatin from formaldehyde-fixed cells was fragmented to a size range of 200–700 bases with a Branson 250 sonifier. Solubilized chromatin was immunoprecipitated with the indicated antibodies overnight at 4°C. Antibody-chromatin complexes were pulled down with protein G-Dynabeads (Life Technologies), washed, and then eluted. After cross-link reversal, RNase A, and proteinase K treatment, immunoprecipitated DNA was extracted with AMP Pure beads (Beckman Coulter). ChIP DNA was quantified with Qubit. 1–5 ng ChIP DNA samples were used to prepare sequencing libraries, and ChIP DNA and input controls were sequenced with the Nextseq 500 Illumina genome analyzer.

Antibodies used for these studies were SMARCA2/4 (39805; Active Motif), H3K4me1 (ab8895; Abcam), H3K4me3 (07-473; Millipore), H3K9ac (ab4441; Abcam), H3K27ac (39133; Active Motif), H3K27me3 (07-449; Millipore), H3K36me3 (ab9050; Abcam), V5 (ab15828; Abcam), RING1B (5694; Cell Signaling), H2AK119ub (8240; Cell Signaling), RYBP (59451204; Sigma-Aldrich), and USP7 (A300-033A; Bethyl Laboratories).

## ChIP-seq data processing and analysis

### Alignment of ChIP-seq reads
Reads were aligned to hg19 (human samples) or mm10 (C3H10T1/2 cells) reference genomes using BWA (Li & Durbin, 2009). Aligned reads were then filtered to exclude PCR duplicates and were extended to 200 bp to approximate fragment sizes. Density maps were generated by counting the number of fragments overlapping each genomic position using igvtools (Thorvaldsdottir et al, 2013), and normalized to 10 million reads. Average ChIP-seq signals across genomic intervals were calculated using bwtool (Pohl & Beato, 2014).

### ChIP-seq peak calling
We used MACS2 (Zhang et al, 2008) to call peaks using matching input controls with a q-value threshold of $10^{-4}$ for SMARCA2/4 and RING1B, and $5 \times 10^{-2}$ for H3K27me3 and using the broad parameter. Peaks within 200 bp of each other were merged and filtered to exclude blacklisted regions as defined by the ENCODE consortium (ENCODE Project Consortium, 2012).

### ChIP-seq peak intersections
Consensus peaks in synovial sarcoma organoids were identified using bedtools (Quinlan & Hall, 2010) and defined as those common to three out of four samples before merging peaks within 2 kb (SMARCA2/4) or 5 kb (H3K27me3) using the Bioconductor genomic ranges package (Lawrence et al, 2013). In C3H10T1/2 cells, SS18-SSX consensus peaks were defined as those common to both samples and with at least twofold more ChIP-seq signals in SS18-SSX samples than in controls.

### ChIP-seq peak annotation
Peaks within 1 kb of RefSeq transcription start sites or with strong H3K4me3 ChIP-seq signals (average normalized signal above eight) were considered as promoters, peaks overlapping a RefSeq gene body were considered as distal intragenic and the remaining peaks were considered distal intergenic. In C3H10T1/2 cells, SS18-SSX consensus peaks were further annotated using the list of known CpG in the mm10 genome (http://hgdownload.cse.ucsc.edu/goldenpath/mm10/database/cpgIslandExt.txt.gz).

### Heat maps and composite plots
Signals shown in heat maps (100 bp windows) and composite plots (10 bp windows) were calculated using bwtool (Pohl & Beato, 2014). Heat map signals (Figs 2D, 4C and H, 6B and I, and 7B) are in $\log_2$ scale, centered on the indicated peaks, and are capped at the 99th percentile. In Figs 2C and 4E, heat maps show average ChIP-seq signals at indicated regions. In Fig 4E, SS18-SSX–binding regions were grouped by hierarchical clustering based on ChIP-seq signals of histone modifications in control samples. Manual curation was used to merge clusters into a reduced number of biologically relevant patterns.

## RNA-seq

Total RNA was isolated from cells using NucleoSpin RNA Plus (Clontech). 0.5–1 μg of total RNA was treated with Ribogold zero to remove ribosomal RNA. Illumina sequencing libraries were constructed using random primers according to the manufacturer's instructions using the Truseq Stranded RNA LT Kit.

### RNA-seq data processing and analysis
Reads were aligned to hg19 or mm10 reference genomes using STAR (Dobin et al, 2013). Mapped reads were filtered to exclude PCR duplicates and reads mapping to known ribosomal RNA coordinates, obtained from rmsk table in the UCSC database (http://genome.ucsc.edu). Gene expression was calculated using featureCounts (Liao et al, 2014). Only primary alignments with mapping quality of 10 or more were counted. Counts were transformed to transcript per million. Signal tracks were generated using bedtools

(Quinlan & Hall, 2010). Gene and transcript abundances were also calculated using Cufflinks (Trapnell et al, 2010).

### Differential expression analysis

Genes differentially expressed between synovial sarcoma organoids and MSCs (Figs 3A and S3C) were identified using DESeq2 (Love et al, 2014) with at least a fourfold change and an adjusted *P*-value < 0.05. For the TCGA sarcoma expression dataset (Cancer Genome Atlas Research Network. Electronic address edsc, Cancer Genome Atlas Research Network, 2017), gene expression levels were obtained by combining all transcripts levels of a given gene. Differentially expressed genes between synovial sarcoma and all other sarcomas (Figs 3D and S3E) and between synovial sarcoma and all other sarcomas taken individually (Fig S3G) were selected based on 5-fold and 20-fold changes in average expression, respectively. Differentially expressed genes in C3H10T1/2 cells upon SS18-SSX lentiviral induction were identified using DESeq2 (Love et al, 2014), with at least fourfold change and an adjusted *P*-value < 0.05.

### Mapping BAF complex peaks to genes

SMARCA2/4 consensus peaks in synovial organoids were mapped to all overlapping (for promoter and distal intragenic peaks) or nearest (for distal intergenic peaks) genes (Fig S3A and B) and attributed to the category of genes expressed at high or low levels in synovial sarcoma. Peak width (Figs 3B and E and S3D) and the percentage of genes associated with narrow or broad peaks (Figs 3C and F and S3F) were then displayed. Similarly, in Fig S3D, SMARCA2/4 peaks in MSCs were mapped to all overlapping or nearest genes, attributed to the category of genes expressed at high or low levels in synovial sarcoma and peak width was then displayed. A similar mapping was performed for SS18-SSX peaks in mouse C3H10T1/2 cells with genes significantly up- or down-regulated upon SS18-SSX expression (Fig 5B).

### Quantification and statistical analysis

All statistical details of experiments are included in the Figure legends or the Materials and Methods section.

### Data and software availability

The data accompanying this article have been deposited into GEO under accession number 148724.

### Additional resources

To aid our analysis, we also used a publicly available gene expression dataset from TCGA for a collection of sarcomas available at (https://toil.xenahubs.net/download/tcga_Kallisto_tpm.gz) (Cancer Genome Atlas Research Network. Electronic address edsc, Cancer Genome Atlas Research Network, 2017).

### Contact for reagent and resource sharing

Further information and requests for resources and reagents should be directed to and will be fulfilled by the Lead Contacts Ivan Stamenkovic (Ivan.Stamenkovic@chuv.ch) and Nicolò Riggi (Nicolo.Riggi@chuv.ch).

## Supplementary Information

## Acknowledgements

This work was supported by Swiss National Science Foundation grant 310030_169563, Switzerland, and Swiss Cancer League grant KLS-4249-08-2017, Switzerland (to I Stamenkovic); and the Swiss National Science Foundation grants PP00P3_157468 and PP00P3_183724, Switzerland, and Swiss Cancer League grant KFS-3973-08-2016, Switzerland, and a Force Foundation grant, Switzerland (to N Riggi). MN Rivera is supported by the Massachusetts General Hospital Research Scholars Program. We gratefully acknowledge Prof Ana Banito (German Cancer Research Center, Heidelberg, Germany) for providing the HYSSII SyS cell line expressing the HA-tagged endogenous SS18-SSX fusion gene.

### Author Contributions

G Boulay: methodology.
L Cironi: methodology.
SP Garcia: investigation.
S Rengarajan: investigation.
Y-H Xing: investigation.
L Lee: investigation.
ME Awad: investigation.
B Naigles: investigation.
S Iyer: formal analysis.
LC Broye: investigation.
T Keskin: investigation.
A Cauderay: investigation.
C Fusco: investigation.
I Letovanec: resources.
I Chebib: resources.
PG Nielsen: resources.
S Tercier: resources.
S Cherix: resources.
T Nguyen-Ngoc: resources.
G Cote: resources.
E Choy: resources.
P Provero: formal analysis.
M-L Suvà: conceptualization and resources.
MN Rivera: conceptualization, funding acquisition, and writing—review and editing.
I Stamenkovic: conceptualization, funding acquisition, and writing—original draft, review, and editing.
N Riggi: conceptualization, funding acquisition, and writing—original draft, review, and editing.

### Conflict of Interest Statement

The authors declare that they have no conflict of interest.

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
