## [Reviewer comments · Life Science Alliance]

Life Science Alliance

The chromatin landscape of primary synovial sarcoma organoids is linked to specific epigenetic mechanisms and dependencies

Gaylor Boulay, Luisa Cironi, Sara Garcia, Shruthi Rengarajan, Yu-Hang Xing, Lukuo Lee, Mary Awad, Beverly Naigles, Sowmya Iyer, Liliane Broye, Tugba Keskin, Alexandra Cauderay, Carlo Fusco, Igor Letovanec, Ivan Chebib, Petur Nielsen, Stephane Tercier, Stephane Cherix, Nguyen-ngoc Tu, Gregory Cote, Edwin Choy, Paolo Provero, Mario Suvà, Miguel Rivera, Ivan Stamenkovic, and Nicolo Riggi

DOI: <https://doi.org/10.26508/lsa.202000808>

Corresponding author(s): Ivan Stamenkovic, University of Lausanne and Nicolo Riggi, University of Lausanne

Review Timeline:

Submission Date:	2020-06-05
Editorial Decision:	2020-06-05
Revision Received:	2020-09-15
Editorial Decision:	2020-11-04
Revision Received:	2020-12-01
Accepted:	2020-12-10

Scientific Editor: Shachi Bhatt

Transaction Report:

Please note that the manuscript was previously reviewed at another journal and the reports were taken into account in the decision-making process at Life Science Alliance. Since the original reviews are not subject to Life Science Alliance's transparent review process policy, the reports and author response cannot be published.

June 5, 2020

Re: Life Science Alliance manuscript #LSA-2020-00808-T

Prof. Ivan Stamenkovic
University of Lausanne
25 Rue du Bugnon, (CHUV)
Lausanne CH-1011

Dear Dr. Stamenkovic,

Thank you for submitting your manuscript entitled "The chromatin landscape of primary synovial sarcoma organoids is linked to specific epigenetic..." to Life Science Alliance. The manuscript was assessed by expert reviewers at another journal before, and the editors transferred those reports to us with your permission.

We can offer further consideration of this manuscript and based on the reports at hand at Life Science Alliance. We would expect a full point-by-point response to reviewer #1's concerns. More specifically:

-Point 1: can get addressed by text changes in the manuscript

-Point 2: address

-Point 3: address by providing support for ncPRC1 contributing to sustaining the transcriptional activation.

-Point 4: analyse the response to USP7 inhibitors in isogenic cells; add the requested discussion points

-Point 5. tone-down conclusions deriving from PLA experiments

- Address other points

We realize that such a revision is demanding and would also necessitate re-review.

We would of course explain the transfer situation and the revision requirements to the reviewer / arbitrating reviewer in this case.

Thank you for this interesting contribution to Life Science Alliance. We are looking forward to receiving your revised manuscript.

Sincerely,

Reilly Lorenz
Editorial Office Life Science Alliance
Meyerhofstr. 1
69117 Heidelberg, Germany
t +49 6221 8891 414
e contact@life-science-alliance.org
www.life-science-alliance.org

B. MANUSCRIPT ORGANIZATION AND FORMATTING:

November 4, 2020

RE: Life Science Alliance Manuscript #LSA-2020-00808-TR

Prof. Ivan Stamenkovic
University of Lausanne
25 Rue du Bugnon, (CHUV)
Lausanne CH-1011
Switzerland

Dear Dr. Stamenkovic,

Thank you for submitting your revised manuscript entitled "The chromatin landscape of primary synovial sarcoma organoids is linked to specific epigenetic..." to Life Science Alliance (LSA), and for addressing the points raised by LSA editors based on the review comments from the previous journal. The revised manuscript and the point-by-point rebuttal has been reviewed by one of our expert advisors, and we would be happy to publish your paper in LSA provided the following points are addressed in the final revised version, and pending final revisions necessary to meet our formatting guidelines (points listed at the end of the this decision letter.

- + please explicitly compare the SyS organoid models (in figs 1-3) with the previously characterized cell culture models
- + please include the EZH2 inhibition data that is right now in the rebuttal letter only, as a Supplementary figure in the manuscript
- + please revised the line in the Discussion 'While our observations suggest that SS18-SSX recruits the ncPRC1.1 complex containing RING1B, RYBP...' given that depletion of RYBP does not lead to the reduction in transcriptional changes induced by SS18
- + please double-check the author list in the manuscript and in our system to be sure that these match
- + please add ORCID ID's for corresponding author and corresponding author
- + please add scale bars to Figure 1A
- + please provide original unprocessed gels for Figures S6A, S7A and S7D

A. FINAL FILES:

B. MANUSCRIPT ORGANIZATION AND FORMATTING:

Sincerely,

Shachi Bhatt, Ph.D.
Executive Editor

Life Science Alliance
<https://www.lsjournal.org/>
Tweet @SciBhatt @LSAJournal

December 10, 2020

RE: Life Science Alliance Manuscript #LSA-2020-00808-TRR

Prof. Ivan Stamenkovic
University of Lausanne
25 Rue du Bugnon, (CHUV)
Lausanne CH-1011
Switzerland

Dear Dr. Stamenkovic,

Thank you for submitting your Research Article entitled "The chromatin landscape of primary synovial sarcoma organoids is linked to specific epigenetic...". It is a pleasure to let you know that your manuscript is now accepted for publication in Life Science Alliance. Congratulations on this interesting work.

DISTRIBUTION OF MATERIALS:

Again, congratulations on a very nice paper. I hope you found the review process to be constructive and are pleased with how the manuscript was handled editorially. We look forward to future exciting submissions from your lab.

Sincerely,

Shachi Bhatt, Ph.D.

Executive Editor

Life Science Alliance

<https://www.lsjournal.org/>
